# Demystifying Catastrophic Forgetting in Two-Stage Incremental Object Detector

Qirui Wu [1]   Shizhou Zhang [1]   De Cheng [2]   Yinghui Xing [1]   Di Xu [3]   Peng Wang [1]   Yanning Zhang [1]

## Abstract

Catastrophic forgetting is a critical chanllenge for incremental object detection (IOD). Most existing methods treat the detector monolithically, relying on instance replay or knowledge distillation without analyzing component-specific forgetting. Through dissection of Faster R-CNN, we reveal a key insight: *Catastrophic forgetting is predominantly localized to the RoI Head classifier, while regressors retain robustness across incremental stages.* This finding challenges conventional assumptions, motivating us to develop a framework termed NSGP-RePRE. Regional Prototype Replay (RePRE) mitigates classifier forgetting via replay of two types of prototypes: coarse prototypes represent class-wise semantic centers of RoI features, while fine-grained prototypes model intra-class variations. Null Space Gradient Projection (NSGP) is further introduced to eliminate prototype-feature misalignment by updating the feature extractor in directions orthogonal to subspace of old inputs via gradient projection, aligning RePRE with incremental learning dynamics. Our simple yet effective design allows NSGP-RePRE to achieve state-of-the-art performance on the Pascal VOC and MS COCO datasets under various settings. Our work not only advances IOD methodology but also provide pivotal insights for catastrophic forgetting mitigation in IOD. Code is available at https://github.com/fanrena/NSGP-RePRE .

## 1. Introduction

As one of the most fundamental tasks in computer vision, significant progress has been made in the field of object detection (Ren et al., 2016; Khanam & Hussain, 2024; Li et al., 2023). Traditional methods mostly solve the object detection task under a static closed-world setting, where all to-be-detected object classes and annotations are fully available before training. Nevertheless, real-world applications frequently encompass dynamic environments where new object categories appear progressively over time. Detectors should possess the capability to adjust to novel tasks through sequential learning, while simultaneously preserving the knowledge gained from detecting previous classes.

Conventional object detectors (Ren et al., 2016; Carion et al., 2020; Li et al., 2020) often suffer from catastrophic forgetting during incremental learning, which significantly hampers their performance in previously learned classes when new tasks are introduced. Unlike incremental learning in classification tasks, incremental object detection (IOD) is more challenging than classification as it requires the simultaneous classification and location of a set of objects in the image. To obtain an incremental object detector with excellent performance, many research efforts have been devoted by introducing knowledge distillation or data replay techniques (Cermelli et al., 2022; Yuyang et al., 2023; Mo et al., 2024) into popular detection frameworks.

Current research in the IOD field usually treats the detector as a whole, and few works pay attention to whether catastrophic forgetting mainly comes from a certain component or whether all modules contribute roughly the same. Demystifying catastrophic forgetting in a sophisticated detector is necessary and helpful not only to establish a bridge between incremental learning in classification and object detection, but also to provide principled guidance for designing simpler and more effective IOD methods. In this study, we chose the widely adopted two-stage Faster R-CNN (Ren et al., 2016) detector as a pioneering research object.

Faster R-CNN is composed of a backbone, neck, region proposal network (RPN) and region of interest head (RoI Head), each of which is crucial to the detector's performance. Our primary focus is on the RPN and RoI Head, known for their key roles in object detection. Through a systematic analysis of its core components, we uncover several important insights. 1) RPN's recall ability remains consistent when transitioning to new tasks. 2) RPN's forgetting has an insignificant impact on overall performance. 3) Forgetting

[1] School of Computer Science, Northwestern Polytechnical University, Xi'an, China [2] School of Telecommunication and Engineering, Xidian University, Xi'an, China [3] Huawei Technologies Ltd. Correspondence to: Shizhou Zhang <szzhang@nwpu.edu.cn>, De Cheng <dcheng@xidian.edu.cn>.

*Proceedings of the 42nd International Conference on Machine Learning*, Vancouver, Canada. PMLR 267, 2025. Copyright 2025 by the author(s).

mainly occurs in the RoI Head's classifier, while the regressor component efficiently retains its knowledge. These findings challenge conventional assumptions and inspire us to propose a novel simple yet effective IOD framework.

In this paper, we propose NSGP-RePRE, which is composed of two components: Regional Prototype REplay (RePRE) and Null Space Gradient Projection (NSGP). To address catastrophic forgetting in the RoI Head classifier, RePRE alleviates forgetting by replaying stored regional prototypes, including coarse regional prototypes and fine-grained regional prototypes of each class. Coarse prototypes act as stable semantic centers, representing the core structure of the RoI feature space. Fine-grained prototypes complement these as a semantic augmentation by capturing intra-class diversity, ensuring a more holistic modeling of the feature distribution. By jointly leveraging these components, RePRE strengthens the capacity of the RoI Head classifier to retain learned knowledge across tasks while accommodating new knowledge, significantly improving incremental learning performance. To prevent toxic replay with misaligned prototypes due to the drift of RoI features caused by updates to the feature extractor, we introduce NSGP to manipulate the gradient in the feature extractor. By projecting gradients into the null space of previous examples, RoI feature distortion is greatly minimized, ensuring prototype and RoI feature alignment. Our approach achieves state-of-the-art results on the PASCAL VOC and COCO datasets under various single and multi-step settings.

Our main contributions are three-fold:

- We comprehensively studied the key components of Faster R-CNN and identified RoI Head classifier as the primary cause of catastrophic forgetting, providing principled guidance for IOD method design.

- Based on our finding, we propose NSGP-RePRE to alleviate forgetting of RoI Head classifier by Regional Prototype Replay complemented with Null Space Gradient Projection for RoI feature anti-drifting.

- Our method not only achieves state-of-the-art performance across multiple datasets under various single and multi-step settings, but also provides pivotal insights for mitigating forgetting in IOD.

## 2. Related Work

Incremental learning, or continual learning, progressively learn new knowledge while retaining previous information. It is categorized into task-incremental, class-incremental, and domain-incremental challenges. The most challenging class-incremental learning is the primary focus of this paper.

### 2.1. Incremental Learning for Classification

Most influential incremental learning studies have focused on classification tasks. Some regularization-based methods enforce the stability of logits (Li & Hoiem, 2017; Zhang et al., 2023; Yan et al., 2025) or intermediate features (Simon et al., 2021) to preserve the learned knowledge, while others apply restrictions on the weight of the model (Kirkpatrick et al., 2017) or on gradients during optimization (Lopez-Paz & Ranzato, 2017; Wang et al., 2021). Structure-based methods are dedicated to learning specific parameters for different tasks, with a dynamically expanding architecture (Rusu et al., 2016) or grouped parameters in a static model (Fernando et al., 2017). For replay-based methods, they can be divided into experience replay (Buzzega et al., 2020; Zhu et al., 2021; Kong et al., 2023) and generative replay methods (Zhai et al., 2020; Kemker & Kanan, 2017), depending on the examples stored in a buffer or generated with a model. Recently, incremental learning based on foundation models such as CLIP (Radford et al., 2021) has also attracted attention. Research works such as L2P (Wang et al., 2022), O-LoRA (Wang et al., 2023a), and VPT-NSP$^2$ (Lu et al.) attempt to learn continuously based on the parameter-efficient transfer learning technique (Zhou et al., 2022; Jia et al., 2022; Xing et al., 2023; Zhang et al., 2025) have achieved superior performance.

### 2.2. Incremental Learning for Object Detection

Incremental object detection presents unique challenges compared with the classification task. IOD are required to locate and classify the visual objects in images. It also faces a distinctive missing annotation problem where potential instances not belonging to the classes of the current learning stage are labeled as background. Most existing IOD works can be summarized into two categories. One is knowledge distillation based methods (Mo et al., 2024; Cermelli et al., 2022). BPF (Mo et al., 2024) bridges past and future with pseudo-labeling and potential object estimation to align models across stages, ensuring a consistent optimization direction. MMA (Cermelli et al., 2022) consolidates the background and all old classes into one entity to minimize the conflict between optimization objects between previous and current tasks. The other is to preserve knowledge through a replay of previous data stored in images (Liu et al., 2023), instances (Yuyang et al., 2023), or features (Acharya et al., 2020). ABR (Yuyang et al., 2023) replayed foreground objects from previous tasks stored in a buffer to reinforce the learned knowledge. RODEO (Acharya et al., 2020) stored compressed representations in a fixed-capacity memory buffer to incrementally perform object detection in a streaming fashion. Unlike existing methods, we delve into analyzing where the forgetting originated for the two-stage incremental object detector. Then we tailor a method specifically designed to combat the crux forgetting module, *i.e.*

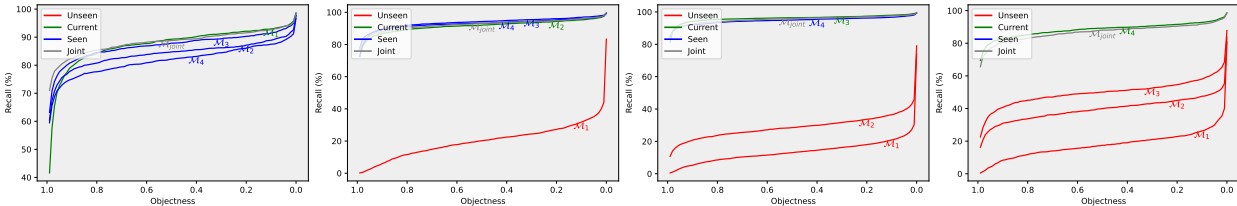

(a) RPN's recall on $\mathcal{D}_1^{test}$.     (b) RPN's recall on $\mathcal{D}_2^{test}$.     (c) RPN's recall on $\mathcal{D}_3^{test}$.     (d) RPN's recall on $\mathcal{D}_4^{test}$.

Figure 1: Recall-Objectness curve of RPN's prediction. IoU threshold is set to 0.5. Blue: $\mathcal{M}_j$ has been trained with training images of $\mathcal{D}_i$ in earlier stages. Green: $\mathcal{M}_j$ is just fine-tuned on $\mathcal{D}_i$. Red: $\mathcal{M}_j$ has not seen the training set of $\mathcal{D}_i$ before. Gray: $\mathcal{M}_{joint}$ is trained jointly on all training images of $\mathcal{D}$.

RoI Head classifier, by replaying RoI features from previously seen tasks to preserve the classification performance.

## 3. Anatomy of Faster R-CNN

### 3.1. Preliminary

**Problem Formulation of Incremental Object Detection.** In Incremental Object Detection, training is structured across n sequential learning stages, with each stage incorporating a new set of classes to be detected. Let $\mathcal{C} = \{\mathcal{C}_1, \mathcal{C}_2, \ldots, \mathcal{C}_t, \ldots, \mathcal{C}_n\}$ represent the entire class set that the detector $\mathcal{M}$ incrementally acquires, with $\mathcal{C}_i \cap \mathcal{C}_j = \emptyset$ for all $i \neq j$. The dataset $\mathcal{D}_t = \{\mathcal{X}_t, \mathcal{Y}_t\}$ comprises images and annotations for the $t$-th learning stage. Each image in $\mathcal{X}_t$ could feature multiple objects of various classes from $\mathcal{C}$, though only those in $\mathcal{C}_t$ are annotated. The main challenge in IOD is to update the detector from $\mathcal{M}_{t-1}$ to $\mathcal{M}_t$ using $\mathcal{D}_t$ solely, without access to earlier datasets $\{\mathcal{D}_1, \ldots, \mathcal{D}_{t-1}\}$, while preserving or enhancing the detector's performance on previously learned classes $\{\mathcal{C}_1, \ldots, \mathcal{C}_{t-1}\}$.

**Faster R-CNN Architecture.** Our study utilizes the two-stage object detector Faster R-CNN, which involves four primary components: a backbone network $f_b$, a neck $f_n$, a Region Proposal Network (RPN) $f_{\text{RPN}}$, and a Region of Interest (RoI) Head $f_{\text{RoI}}$. The backbone and neck modules are responsible for feature extraction, and their combination is represented as $f_{nb} = f_n \circ f_b$. The RPN generates object proposal boxes accompanied by objectness scores, which express the likelihood of each box containing a target object. Following this, proposals with higher objectness scores $\mathbf{P}$ are chosen for RoI feature extraction using RoI Align. The RoI Head is divided into two branches: the classification branch $f_{cls}$ and the regression branch $f_{bbox}$. The obtained RoI features $\mathbf{P}$ are fed into these branches to classify and adjust the positions of the bounding boxes.

### 3.2. Rationale for Anatomy

When adapting Faster R-CNN to sequential learning tasks, catastrophic forgetting is the primary limitation. The central question driving this work is: *Which component predominantly leads to catastrophic forgetting, or do all components have a contributing role?* To systematically address this, we decompose the ultimate question into three interconnected sub-questions: 1. Can RPN retain its recall ability in incremental learning? RPN functions as an initial object localizer and its recall rate plays a critical role in the overall performance of the detector. 2. How much does RPN's forgetting affect the final performance of the detector? RPN doesn't produce final predictions on classification nor localization, it is crucial to investigate the actual impact caused by its degradation. 3. Which branch of the RoI Head predominantly accounts for forgetting? The RoI Head is responsible for the ultimate prediction of the detector, its dual role in classification and modifying the bounding box potentially makes it sensitive to task-specific changes. To answer these questions, we conduct a series of analytical experiments in the following section from a statistical perspective, as the detector learns sequentially.

Our analytical experiments are conducted on the PASCAL VOC dataset, starting with five classes and incrementally adding five classes across three additional stages. We employ pseudo-labeling as a basic strategy to mitigate the missing annotation issue and top 1,000 proposals are selected to provide a sufficient number for our investigation. In the following sections, we evaluate the RPN and RoI Head within the detectors learned on all stages and a jointly trained detector, *i.e.* $\mathcal{M}_1$ to $\mathcal{M}_4$ and $\mathcal{M}_{joint}$, on the test set of each learning stage, *i.e.* $\mathcal{D}_1^{test}$ to $\mathcal{D}_4^{test}$.

To clarify, we use various colors to depict the performance of Model $\mathcal{M}_t$ on test set of $\mathcal{D}_i^{test}$ across different training stages. Green indicates $\mathcal{M}_t$ when it is just fine-tuned in its corresponding stage with the training set of $\mathcal{D}_i$ ($t = i$), displaying peak performance on test set of $\mathcal{D}_i^{test}$. Blue represents $\mathcal{M}_t$ that has encountered training set of $\mathcal{D}_i$ in earlier stages ($t > i$), demonstrating the phenomenon of forgetting after multiple training stages. Red illustrates $\mathcal{M}_t$ that has not been fed with training images of $\mathcal{D}_i$ before ($t < i$), highlighting the model's generalization capability.

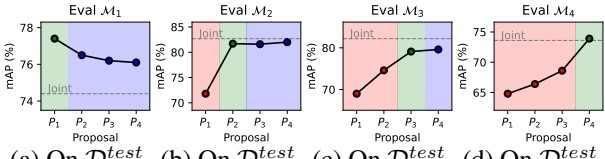

(a) On $\mathcal{D}_1^{test}$  (b) On $\mathcal{D}_2^{test}$  (c) On $\mathcal{D}_3^{test}$  (d) On $\mathcal{D}_4^{test}$

Figure 2: Results of $\mathcal{M}_i$ on $\mathcal{D}_i$ with different proposals. $\mathbf{P}_j$ are produced by corresponding $\mathcal{M}_j$.

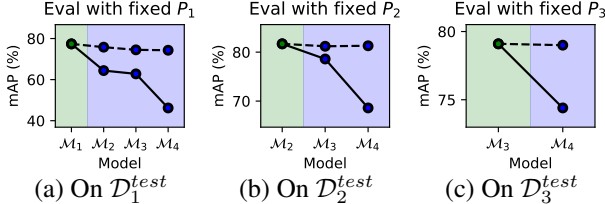

(a) On $\mathcal{D}_1^{test}$  (b) On $\mathcal{D}_2^{test}$  (c) On $\mathcal{D}_3^{test}$

Figure 3: Results of $\mathcal{M}_i$ on various $\mathcal{D}_i$ by using a fixed set of proposals. "- -" indicates the classification results of each proposal is designated by Model freshly trained on the corresponding $\mathcal{D}$. "—" indicates the predicted classification results for the corresponding model in the x-axis.

### 3.3. Anatomy of Faster R-CNN

**RPN's recall ability remains consistent across sequential tasks.** RPN allows the detector to generate possible RoIs and deterioration in proposal quality will hamper the detector's final performance. It is essential to examine the RPN's recall rate from a statistical view, as it can reflect the knowledge-preserving ability of RPN. We perform experiments on the RPNs within the detectors learned on all the four stages and the jointly learned detector, *i.e.* $\mathcal{M}_1$ to $\mathcal{M}_4$ and $\mathcal{M}_{joint}$, by using Recall-Objectness curves on all the four test sets of each training stage $\mathcal{D}_1^{test}$ to $\mathcal{D}_4^{test}$. Note that the threshold of IoU between the proposals and GTs is set to 0.5. As shown in Figure 1 (a), the blue curves represent the recall ability of RPNs within $\mathcal{M}_2$ to $\mathcal{M}_4$ which have been previously tuned on training set of $\mathcal{D}_1$. The green curve shows the recall of $\mathcal{M}_1$, which has just been fine-tuned on $\mathcal{D}_1$. It can be clearly seen that the blue curves show a slight reduction compared to the green curve. As the objectness score approaches 0, the recall rates of various models improve to similar outcomes close to 100%. (Note that only top 1,000 proposals are selected in our experiments.) The slight reduction between the blue and green curves highlights that the RPN experiences little forgetting after multiple sequential learning stages. This trend is also observed in Figure 1 (b) and (c).

**RPN's minimal forgetting negligibly affects overall performance.** To assess the actual impact of RPN's forgetting on the detector's final performance, we adopt proposals generated from models of subsequent training stages ($\mathcal{M}_{i+1}$ to $\mathcal{M}_n$) to the current model on the current stage $\mathcal{M}_i$, to test the final performance of $\mathcal{M}_i$. $\mathbf{P}_i$ denotes the propos-

als generated by $\mathcal{M}_i$. As depicted in Figure 2 (a), $\mathcal{M}_1$ is evaluated on test set of $\mathcal{D}_1^{test}$ with varying sets of proposals. $\mathbf{P}_1$ demonstrates the optimal performance of $\mathcal{M}_1$ since $\mathbf{P}_1$ is generated with the RPN of $\mathcal{M}_1$, *i.e.* zero forgetting. Although $\mathbf{P}_2$ to $\mathbf{P}_4$ exhibit some forgetting compared to $\mathbf{P}_1$, the performance deterioration of $\mathcal{M}_1$ with $\mathbf{P}_2$ to $\mathbf{P}_4$ is minimal, with only a 1.3% reduction in mAP observed on $\mathcal{D}_1$, from $\mathbf{P}_1$ (77.4%) to $\mathbf{P}_4$ (76.1%). This minor decrease suggests that RPN's forgetting has minor impact on the detector's final performance. Consistent conclusion can be obtained from Figure 2 (b) and (c).

**The RoI Head classifier exhibits severe catastrophic forgetting.** As discussed previously, RPN contributes minimally to the detector's forgetting. To investigate the crux of forgetting, we fixed the proposals $\mathbf{P}_i$ generated with $\mathcal{M}_i$ and fed them into RoI Heads of detectors in subsequent stages $\mathcal{M}_{i+1}$ to $\mathcal{M}_n$. By designating the classification results of each proposal with $\mathcal{M}_i$'s results, we can isolate the forgetting caused by the regression branch and the classification branch. As shown in Figure 3 (a), the dashed line represents the mAP of models designated with the $\mathcal{M}_1$'s classification results, while the solid line represents the classification results produced by the corresponding models on the x-axis. In Figure 3 (a), the dashed line remains almost unchanged, suggesting the forgetting caused by the regressor is minor. The solid line deteriorates rapidly as more stages have been trained on the detector, indicating that the classification head primarily causes the forgetting. The same trend in Figure 3 (b) and (c) further confirms the conclusion.

Interestingly, our findings also reveal that RPN effectively generalizes to previously unseen classes as can be seen from the red curves shown in Figure 2 and Figure 3. More detailed analysis are presented in the appendix. We note that the reason for minimal forgetting in regression and large forgetting in classifier is unclear. We infer that it may be due to the absence of task conflicts in the detector regression branches, while it is severe in the classification task (Wang et al., 2023b; Huang et al., 2024) for incremental learning.

### 3.4. Key Findings

Through statistical evaluation and systematic analysis, we demonstrate three key findings:

- **RPN Recall Stability:** In sequential tasks, the stability of the RPN recall ability is largely maintained.

- **RPN's Impact on Performance:** RPN's minimal forgetting has a negligible impact on overall performance.

- **RoI Head Classifier Vulnerability:** The RoI Head classifier suffers severely from catastrophic forgetting, while the regressor can efficiently retain its knowledge.

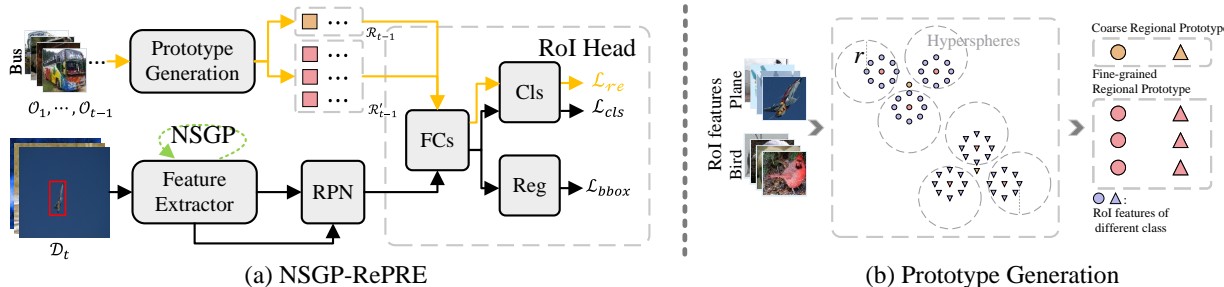

(a) NSGP-RePRE                                        (b) Prototype Generation

Figure 4: The overall architecture of our NSGP-RePRE framework. This framework incorporates RePRE to mitigate forgetting within the RoI Head's classifier. NSGP is introduced to counteract the shifts induced by the evolving feature extractor.

Our analysis reveals that *Catastrophic forgetting in Faster R-CNN stems predominantly from the instability of the RoI Head's classifier, rather than degradation in RPN's recall capability or the regression branch of RoI Head.* Our analyses demonstrate minimal forgetting in regression, building a bridge between classical incremental classification and two-stage incremental object detector. This offers fundamental insights for developing simpler and more efficient IOD methods. Consequently, we present a straightforward and effective approach to address forgetting in the RoI Head classifier, thereby reducing forgetting in the detector.

## 4. Method

**Overview of Framework.** Earlier discussions have pinpointed that the crux of catastrophic forgetting in Faster R-CNN mainly stems from the classification branch of the RoI Head, establishing a bridge between incremental classification and incremental object detection. Based on our previous analytical results, we propose a simple yet effective Regional Prototype Replay (RePRE) incorporated with Null Space Gradient Projection (NSGP) framework termed NSGP-RePRE specifically targeting the forgetting in RoI Head classifier.

As depicted in Figure 4, NSGP-RePRE employs NSGP for regulating the backbone and neck, while RePRE manages the RoI Head. RePRE creates coarse regional prototypes from RoI features of each class, along with fine-grained regional prototypes to enhance semantic diversity. These prototypes are replayed via the RoI Head's classification branch. Unlike previous works (Yuyang et al., 2023; Mo et al., 2024), RePRE provides consistent guidance with minimal prototype storage per class to prevent forgetting specifically on RoI Head classifier. Addressing the issue of prototype-feature misalignment identified in prior research (Yu et al., 2020; Gomez-Villa et al., 2025), NSGP is introduced to restrict changes in RoI features, ensuring prototype's alignment with the evolving RoI feature distributions.

### 4.1. RePRE

RePRE retains the previously learned classification knowledge by replaying regional prototypes from the past. Specifically, to obtain coarse regional prototypes for the RePRE in the next training stage $t + 1$, we extract RoI features $\mathcal{O}_t = \{\mathbf{o}_i^c \mid i \in \mathbb{N}, c \in \mathbb{N}, 1 \leq i \leq n_c, \bar{N}_{t-1} \leq c \leq \bar{N}_t\}$ from the feature maps as

$$\mathbf{o}_i^c = \text{RoIAlign}(\mathbf{P}_i^c, f_{nb}(\mathbf{x}_i^c)), \quad (1)$$

where $\mathbf{P}_i^c$ are the proposals covering class $c$, $\mathbf{x}_i^c$ is the image containing objects of $c$ and $n_c$ is the number of proposals that cover object from class $c$, $\bar{N}_t$ represents the total class number of $\mathcal{C}_{old} = \{\mathcal{C}_1, \cdots, \mathcal{C}_t\}$. The generation of $\mathbf{P}_i^c$ can be expressed as

$$\mathbf{P}_i^c = f_{rpn}(f_{nb}(\mathbf{x}_i^c)). \quad (2)$$

Next, we compute and store a single prototype for each class given by

$$\boldsymbol{\mu}_c = \frac{1}{n_c} \sum_{i=1}^{n_c} \mathbf{o}_i^c. \quad (3)$$

The resulting $\boldsymbol{\mu}_c$ will be appended to $\mathcal{R}_{t-1} = \{\boldsymbol{\mu}_k \mid k \in \mathbb{N}, 1 \leq k \leq \bar{N}_{t-1}\}$ to form $\mathcal{R}_t$, which stores prototypes from past stages.

To capture the entire spectrum of useful information on the distribution of RoI features. We introduce complementary fine-grained regional prototypes chosen through a density-aware prototype selection strategy. Specifically, we first calculate the cosine similarity between RoI features extracted via RoI Align:

$$s_{i,j}^c = \frac{\mathbf{o}_i^c \mathbf{o}_j^c}{||\mathbf{o}_i^c||||\mathbf{o}_j^c||}, 1 \leq i, j \leq n_c \quad (4)$$

For each RoI feature , we define a hypersphere with radius $r$, centered at $\mathbf{o}_j^c$ , that contains neighboring features $\mathcal{S}_j^c = \{\mathbf{o}_i^c \mid s_{i,j}^c > r, 1 \leq i \leq n_c\}$. The importance of a hypersphere is quantified by its cardinality (i.e. the

number of RoI features it contains). To ensure diversity and avoid redundancy, we greedily select the top-$K$ hyperspheres $\{\mathcal{S}_j^c \mid 1 \leq j \leq K\}$ in descending order of importance. During selection, any candidate hypersphere whose center lies within the radius $r$ of a previously selected (more important) hypersphere is excluded. The fine-grained regional prototypes are computed by averaging all features in their corresponding hypersphere:

$$\boldsymbol{\mu}'_{c,j} = \frac{1}{|\mathcal{S}_j^c|} \sum_{\mathbf{o} \in \mathcal{S}_j^c} \mathbf{o}, \tag{5}$$

and these prototypes are added to the fine-grained prototype buffer $\mathcal{R}'_{t-1} = \{\boldsymbol{\mu}'_{k,j} \mid k, j \in \mathbb{N}, 1 \leq k \leq \bar{N}_{t-1}, 1 \leq j \leq K\}$.

To replay these prototypes, at stage $t + 1$, a regional prototype $\boldsymbol{\mu}_k$ is fed into the classification branch of the RoI Head to predict the class probabilities as

$$\hat{\mathbf{y}}_k = \text{Softmax}(f_{cls}(\boldsymbol{\mu}_k)). \tag{6}$$

The replay loss $\mathcal{L}_{re}$ is computed as

$$\mathcal{L}_{re} = - \sum_{\mathbf{y}_k \in C_{old}} \mathbf{y}_k \log \hat{\mathbf{y}}_k - \sum_{\mathbf{y}_k \in C_{old}} \sum_{i=1}^{K} \mathbf{y}_k \log \hat{\mathbf{y}}'_{k,i}, \tag{7}$$

where $\mathbf{y}_k$ represents the ground-truth label associated with the coarse prototype $\boldsymbol{\mu}_k$ and fine-grained prototype $\boldsymbol{\mu}'_{k,i}$. The overall loss function for the detector is then formulated as:

$$\mathcal{L} = \mathcal{L}_{cls} + \mathcal{L}_{bbox} + \mathcal{L}_{re}, \tag{8}$$

where $\mathcal{L}_{cls}$ and $\mathcal{L}_{bbox}$ correspond to the classification and bounding box regression losses for the current stage $t$.

### 4.2. NSGP for RoI Features Anti-drifting

When using regional prototype replays that stabilize the RoI Head, the continuously updating feature extractor may cause the features of previous classes to drift. The drift results in a misalignment between the stored prototypes and the RoI features in the current training stage, which will hamper the model to retain its knowledge. To reduce distortions in the RoI feature space during learning new tasks, we introduce a Null-Space Gradient Projection (NSGP) strategy to prevent updating of the backbone and neck from interfering with the features of previously seen tasks. RePRE and NSGP work together to form an exquisite incremental object detector, with RePRE managing the RoI Head and NSGP governing the backbone and neck.

Denote the parameters of the Convolution/FC layer in the backbone and neck as $\mathbf{W}$, and the gradient $\mathbf{G}$ is calculated by the backward pass. To ensure that updating based on $\mathbf{G}$ will not change previous tasks, NSGP projects $\mathbf{G}$ into

the null space of the previous samples (Wang et al., 2021), to obtain $\Delta \mathbf{W}$. This projection ensures that $\Delta \mathbf{W}$ remains orthogonal to the inputs of the old tasks $\mathcal{X}$. Consequently, the update can be formulated as

$$\mathbf{W}_{t+1} = \mathbf{W}_t - \alpha \Delta \mathbf{W}_t, \tag{9}$$

in the time step $t$, where $\alpha$ is learning rate. The orthogonality condition between $\mathcal{X}$ and $\Delta \mathbf{W}_t$ ensures

$$\mathcal{X}(\mathbf{W}_t - \alpha \Delta \mathbf{W}_t) = \mathcal{X}\mathbf{W}_t \tag{10}$$

is satisfied, effectively preventing drifts in the feature extractor's updates. We adjusted the projection matrix in NSGP to enhance its compatibility with Faster R-CNN. Additional details can be found in the Appendix.

In general, the NSGP will control the $\mathbf{G}$ of the backbone and neck, ensuring that they are projected into the null space corresponding to input from previous examples. This approach stabilizes the RoI features, thus improving not only the classification accuracy but also the minimal forgetting in regression.

## 5. Experiments

### 5.1. Experimental Settings

**Datasets and Evaluation Metrics.** Following the same protocols as in previous works (Yuyang et al., 2023; Mo et al., 2024), we evaluate our method on the PASCAL VOC 2007 (Everingham et al., 2010) and MS COCO 2017 (Lin et al., 2014) datasets. PASCAL VOC 2007 contains 20 different classes, including 9,963 annotated images. MS COCO 2017 dataset comprises 80 classes, with around 118k images for training and 5,000 images for validation. The mean average precision at the 0.5 IoU threshold (mAP@0.5) is used as the primary evaluation metric for VOC dataset, and the mean average precision ranging from 0.5 to 0.95 is the main evaluation metric for the COCO dataset. For each incremental setting (A-B), the first number A denotes the number of classes in the first task, while the second number B represents the number of classes in the subsequent tasks.

**Implementation Details.** Similar to previous works (Yuyang et al., 2023; Mo et al., 2024), we build our incremental Faster R-CNN (Ren et al., 2016) with R50 (He et al., 2016). In our method, we incorporate a pseudo-labeling strategy to solve the missing annotation problem as in BPF (Mo et al., 2024). More implementation details can be found in the appendix.

### 5.2. Quantitative Evaluation

Following previous works (Yuyang et al., 2023; Mo et al., 2024), our method is evaluated on various settings including single-step and multi-step increments. We compare

Table 1: mAP@0.5 results on single incremental step on PASCAL VOC 2007. The best performance in each is presented with **bold**, and the second best is presented with underline.

| Method | 19-1 | | | | 15-5 | | | | 10-10 | | | | 5-15 | | | |
|---|---|---|---|---|---|---|---|---|---|---|---|---|---|---|---|---|
| | 1-19 | 20 | 1-20 | Avg | 1-15 | 16-20 | 1-20 | Avg | 1-10 | 11-20 | 1-20 | Avg | 1-5 | 5-15 | 1-20 | Avg |
| Joint | 76.4 | 76.4 | 76.4 | 76.4 | 78.3 | 70.7 | 76.4 | 74.5 | 76.9 | 76.0 | 76.4 | 76.4 | 73.6 | 77.4 | 76.4 | 75.5 |
| Fine-tuning | 12.0 | 62.8 | 14.5 | 37.4 | 14.2 | 59.2 | 25.4 | 36.7 | 9.5 | 62.5 | 36.0 | 36.0 | 6.9 | 63.1 | 49.1 | 35.0 |
| ORE (Joseph et al., 2021a) | 69.4 | 60.1 | 68.9 | 64.7 | 71.8 | 58.7 | 68.5 | 65.2 | 60.4 | 68.8 | 64.6 | 64.6 | - | - | - | - |
| OW-DETR (Gupta et al., 2022) | 70.2 | 62.0 | 69.8 | 66.1 | 72.2 | 59.8 | 69.1 | 66.0 | 63.5 | 67.9 | 65.7 | 65.7 | - | - | - | - |
| ILOD-Meta (Joseph et al., 2021b) | 70.9 | 57.6 | 70.2 | 64.2 | 71.7 | 55.9 | 67.8 | 63.8 | 68.4 | 64.3 | 66.3 | 66.3 | - | - | - | - |
| ABR (Yuyang et al., 2023) | 71.0 | **69.7** | 70.9 | 70.4 | 73.0 | **65.1** | 71.0 | 69.1 | 71.2 | 72.8 | 72.0 | 72.0 | 64.7 | 71.0 | 69.4 | 67.9 |
| FasterILOD (Ren et al., 2016) | 68.9 | 61.1 | 68.5 | 65.0 | 71.6 | 56.9 | 67.9 | 64.3 | 69.8 | 54.5 | 62.1 | 62.1 | 62.0 | 37.1 | 43.3 | 49.6 |
| PPAS (Zhou et al., 2020) | 70.5 | 53.0 | 69.2 | 61.8 | - | - | - | - | 63.5 | 60.0 | 61.8 | 61.8 | - | - | - | - |
| MVC (Yang et al., 2022) | 70.2 | 60.6 | 69.7 | 65.4 | 69.4 | 57.9 | 66.5 | 63.7 | 66.2 | 66.0 | 66.1 | 66.1 | - | - | - | - |
| PROB (Zohar et al., 2023) | 73.9 | 48.5 | 72.6 | 61.5 | 73.5 | 60.8 | 70.1 | 67.0 | 66.0 | 67.2 | 66.5 | 66.5 | - | - | - | - |
| PseudoRM (Yang et al., 2023) | 72.9 | 67.3 | 72.6 | 70.1 | 73.4 | 60.9 | 70.3 | 66.9 | 69.1 | 68.6 | 68.9 | 68.9 | - | - | - | - |
| MMA (Cermelli et al., 2022) | 71.1 | 63.4 | 70.7 | 67.2 | 73.0 | 60.5 | 69.9 | 66.7 | 69.3 | 63.9 | 66.6 | 66.6 | 66.8 | 57.2 | 59.6 | 62.0 |
| BPF (Mo et al., 2024) | 74.5 | 65.3 | 74.1 | 69.9 | 75.9 | 63.0 | 72.7 | 69.5 | 71.7 | 74.0 | 72.9 | 72.9 | 66.4 | 75.3 | 73.0 | 70.9 |
| NSGP-RePRE | 76.3 | 69.0 | 76.0 | 72.7 | 77.5 | 61.8 | 73.6 | 69.7 | 75.3 | 72.7 | 74.0 | 74.0 | 68.5 | 74.5 | 73.0 | 71.5 |

Table 2: mAP@0.5 results on multiple incremental steps on PASCAL VOC 2007. The best performance in each is presented with **bold**, and the second best is presented with underline.

| Method | 10-5(3tasks) | | | 5-5(4tasks) | | | 10-2(6tasks) | | | 15-1(6tasks) | | | 10-1(11tasks) | | |
|---|---|---|---|---|---|---|---|---|---|---|---|---|---|---|---|
| | 1-10 | 11-20 | 1-20 | 1-5 | 6-20 | 1-20 | 1-10 | 11-20 | 1-20 | 1-15 | 16-20 | 1-20 | 1-10 | 11-20 | 1-20 |
| Joint | 76.9 | 76.0 | 76.4 | 73.6 | 77.4 | 76.4 | 76.9 | 76.0 | 76.4 | 78.3 | 70.7 | 76.4 | 76.9 | 76.0 | 76.4 |
| Fine-tuning | 5.3 | 30.6 | 18.0 | 0.5 | 18.3 | 13.8 | 3.8 | 13.6 | 8.7 | 0.0 | 10.5 | 5.3 | 0.0 | 5.1 | 2.6 |
| ABR (Yuyang et al., 2023) | 68.7 | 67.1 | 67.9 | **64.7** | 56.4 | 58.4 | 67.0 | 58.1 | 62.6 | 68.7 | **56.7** | 65.7 | 62.0 | **55.7** | 58.9 |
| FasterILOD (Ren et al., 2016) | 68.3 | 57.9 | 63.1 | 55.7 | 16.0 | 25.9 | 64.2 | 48.6 | 56.4 | 66.9 | 44.5 | 61.3 | 52.9 | 41.5 | 47.2 |
| MMA (Cermelli et al., 2022) | 66.7 | 61.8 | 64.2 | 62.3 | 31.2 | 38.9 | 65.0 | 53.1 | 59.1 | 68.3 | 54.3 | 64.1 | 59.2 | 48.3 | 53.8 |
| BPF (Mo et al., 2024) | 69.1 | 68.2 | 68.7 | 60.6 | 63.1 | 62.5 | 68.7 | 56.3 | 62.5 | 71.5 | 53.1 | 66.9 | 62.2 | 48.3 | 55.2 |
| NSGP-RePRE | **72.4** | 67.6 | **70.0** | 64.6 | 66.1 | 65.7 | 70.1 | 58.8 | 64.4 | 77.7 | 55.0 | 72.0 | 69.9 | 55.1 | 62.5 |

Table 3: mAP results on MS COCO 2017 at different IoU. The best performance in each is presented with **bold**, and the second best is presented with underline.

| Method | 40-40 | | | 70-10 | | |
|---|---|---|---|---|---|---|
| | AP | AP50 | AP75 | AP | AP50 | AP75 |
| Joint | 36.7 | 57.8 | 39.8 | 36.7 | 57.8 | 39.8 |
| Fine-tuning | 19.0 | 31.2 | 20.4 | 5.6 | 8.6 | 6.2 |
| ILOD-Meta (Joseph et al., 2021b) | 23.8 | 40.5 | 24.4 | - | - | - |
| ABR (Yuyang et al., 2023) | 34.5 | 57.8 | 35.2 | 31.1 | 52.9 | 32.7 |
| FasterILOD (Ren et al., 2016) | 20.6 | 40.1 | - | 21.3 | 39.9 | - |
| PseudoRM (Yang et al., 2023) | 25.3 | 44.4 | - | - | - | - |
| MMA (Cermelli et al., 2022) | 33.0 | 56.6 | 34.6 | 30.2 | 52.1 | 31.5 |
| BPF (Mo et al., 2024) | 34.4 | 54.3 | 37.3 | 36.2 | 56.8 | 38.9 |
| NSGP-RePRE | 35.4 | 55.3 | 38.6 | 36.5 | 56.0 | 39.8 |

our method against two baselines: Joint Training, which involves training the model on the complete dataset using all annotations, and Fine-Tunning, where the model is incrementally trained on new data without any regularization strategy or data replay.

### 5.2.1. PASCAL VOC 2007

On the PASCAL VOC 2007 dataset, we assess our approaches using a single-step incremental task setting, which includes 19-1, 15-5, 10-10, and 5-15 tasks. We also examine a multi-step incremental task setting, covering settings such as 10-5, 5-5, 10-2, 15-1, and 10-1.

**Single-step Increments.** In Table 1, we make a comparison between our proposed method and existing approaches. Our method frequently surpasses others in a range of settings, particularly in the base classes in the initial learning stage, demonstrating its superior ability to mitigate catastrophic forgetting. Specifically, NSGP-RePRE exceeds the previous leading replay-based approach ABR, by an average of 4.4% in the initial class set. It also exceeds the previous SOTA method BPF by 2.3% in the initial class set, bolstering our assertion regarding the superior anti-forgetting capability of our approach. NSGP-RePRE exceeds ABR by 3.3% and BPF by 1% in all 20 classes, underscoring the effectiveness of our method. The Avg metric equally average base and new classes mAP, showing stability and plasticity balance without the influence of the number of classes. In Avg, our method surpasses ABR and BPF by 2.1% and 1.2%, respectively, demonstrating that our method prevails in balance between stability and plasticity in all methods.

**Multi-step Increments.** The issue of catastrophic forgetting becomes more challenging in longer incremental settings. As demonstrated in Table 2, fine-tuning nearly completely

Table 4: Ablation study on each component. Where "Coarse" indicates coarse prototypes replay only, "Fine" indicates fine-grained regional prototypes are also incorporated.

| Model | NSGP | Coarse | Fine | VOC(5-5) | | | | |
|---|---|---|---|---|---|---|---|---|
| | | | | 1-5 | 6-10 | 11-15 | 16-20 | 1-20 |
| (a) | | | | 46.6 | 56.5 | 71.1 | 59.6 | 58.4 |
| (b) | ✓ | | | 62.3 | 60.4 | 73.1 | 57.4 | 63.3 |
| (c) | | ✓ | | 49.8 | 61.0 | 73.5 | 60.5 | 61.2 |
| (d) | ✓ | ✓ | | 65.9 | 61.6 | 73.8 | 56.0 | 64.3 |
| (e) | ✓ | ✓ | ✓ | 64.6 | 66.2 | 73.1 | 59.0 | 65.7 |

forgets the initial classes. NSGP-RePRE shows a 4.7% improvement over ABR in initial classes across all 5 settings, and a 4.2% improvement in all 1-20 classes. Our method exceeds the performance of BPF by 4.5% in the base classes and 2.3% in the 1-20 classes. In the particularly demanding 10-1 settings, our method is 3.6% better than ABR, highlighting the efficacy of our proposed approaches. The improvements observed in more complex multi-step increment settings further validate the effectiveness of our proposed methods.

### 5.2.2. MS COCO 2017

On MS COCO 2017 dataset, we performed experiments on 40-40 and 70-10 settings using the same protocol in comparison methods. As shown in Table 3, fine-tuning suffers from catastrophic forgetting in both settings. While previous approaches have been enhanced with fine-tuning, NSGP-RePRE increased the average AP by 1.0% over the previous state-of-the-art in the 40-40 configuration. In the 70-10 scenario, the performance is close to that of joint training, with our method yielding 0.3% improvements over the previous SOTA BPF. These experimental results demonstrate the efficacy of our approach.

### 5.3. Further Analysis

**Effectiveness of Each Component.** In Table 4, we analyze the effectiveness of NSGP, Coarse, and Fine under the VOC 5-5 setting, where "Coarse" indicates that only coarse prototypes are adopted during replay while "Fine" shows the results incorporated with fine-grained regional prototypes. Variant a denotes our baseline model using pseudo-labeling. Variant b denotes that NSGP is employed to solve the feature drift based upon a, which significantly reduces the catastrophic forgetting of old classes, thus markedly improving old class detection over a. Variant c incorporates RePRE with coarse prototypes only to mitigate catastrophic forgetting. However, performance remains suboptimal due to the feature shift from the updating of the feature extractor. The variant d denotes our NSGP-RePRE with coarse prototypes only, which substantially reduces catastrophic forgetting and demonstrates the efficacy of the method. NSGP-RePRE

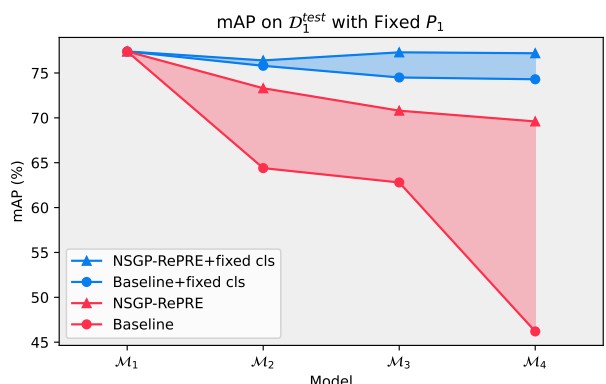

Figure 5: mAP of different model on $\mathcal{D}_1^{test}$ in VOC(5-5) settings. To better demonstrate the impact of our method on the classifier, $P_1$ is fixed to all models. Fixed cls indicates the models classification results is designated by $\mathcal{M}_1$.

achieves the highest performance among all models, exceeding d by 1.4% in the 1-20 division, underscoring the effectiveness of our method. As shown in Table 4, each adopted component independently reduces forgetting and reaches peak performance when used together.

**Anti-forgetting in RoI Head's classifier.** As we intend to minimize the classification error caused by the forgetting in RoI Head's classifier, we demonstrate that our method can effectively solve the problem. As shown in Figure 5, we fixed a set of proposals predicted by the $\mathcal{M}_1$ as the proposals for all $\mathcal{M}$. Fixed cls indicates that the model classification results are designated by $\mathcal{M}_1$. The baseline is the detector only applied with a pseudo-labeling strategy. $\mathcal{M}_1$ is the ideal upper bound in $\mathcal{D}_1^{test}$ as it is freshly trained on $\mathcal{D}_1$. From Figure 5, we can draw some conclusions: 1. By comparing the red curves, we can see that our method has a better classification performance, suggesting the effectiveness of our proposed method. 2. Suggested by the light blue area, NSGD can further reduce the already minimal forgetting in regression. 3. Though the classifier specifically focuses on reducing classification error, the extra components introduced by the method will not disrupt the observation that the regressor exhibits minimal forgetting.

## 6. Conclusion

This study investigates Faster R-CNN as the representative two-stage incremental object detector and demonstrates that catastrophic forgetting primarily originates from the RoI Head's classifier while regressor exhibits minimal forgetting. The finding can provide principled guidelines for designing simple yet effective IOD method. Consequently, we introduce the NSGP-RePRE framework to mitigate forgetting in the RoI Head classifier complemented with NSGP

on the feature extractor. Our extensive experimental results demonstrate the efficacy of the proposed methods. We hope that our research will offer significant insights into IOD, facilitating progress in this area.

## Acknowledgements

This work was supported in part by the National Natural Science Foundation of China (NSFC) under Grant 62476223, 62176198, 62201467, the Key Research and Development Program of Shaanxi Province under Grant 2024GX-YBXM-135, in part by the Young Talent Fund of Xi'an Association for Science and Technology under Grant 959202313088, Innovation Capability Support Program of Shaanxi (No. 2024ZC-KJXX-043).

## Impact Statement

This paper presents work whose goal is to advance the field of Machine Learning. There are many potential societal consequences of our work, none which we feel must be specifically highlighted here.

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

## A. Implementation Details.

Similar to previous works, we use the Faster R-CNN architecture with a Resnet-50 (He et al., 2016) backbone pre-trained in ImageNet (Russakovsky et al., 2015). On PASCAL VOC dataset, we train the network with SGD optimizer, momentum of 0.9 and weight decay of 10e-4. We use a learning rate of 0.02 for all tasks. For MS COCO, we adopt AdamW as the optimizer, weight deacy of 0.01 and learning rate of 5e-5. Batch size is set to 16 for both datasets. In NSGP, we follow the adaptive selecting stategy proposed in (Lu et al.) to keep the singular vaules. We sample 9 extra fine-grained prototypes to complement the coarse prototype, 10 prototypes are used in total. The radius $r$ is set to 0.6. In our method, we incorporate a pseudo-labeling strategy to solve the foreground shift problem as implemented in BPF (Mo et al., 2024). All of our experiments were conducted on 2 RTX 3090 GPU.

## B. Generalization on unseen classes of RPN.

Interestingly, our findings reveal that RPN effectively generalizes to previously unseen classes. As depicted in Figure 1, the red lines represent the RPN's recall for objects belonging to unseen classes. Figure 1 (d) illustrates how RPNs of $\mathcal{M}_1$ to $\mathcal{M}_3$ successfully recall certain objects belonging to classes in the 4-th stage. It can be clearly seen that the recall ability of $\mathcal{M}_1$ to $\mathcal{M}_3$ on test set of $\mathcal{D}_4^{test}$ can be consistently improved after sequential learning. A similar trend is seen in Figure 1 (b) and (c), suggesting RPN's potential to enhance zero-shot detection with sufficient training data. In Figure 2 (d), the red dots represent the outcomes of testing $\mathcal{M}_4$ on the test set of $\mathcal{D}_4^{test}$ employing proposals $\mathbf{P}_1$ to $\mathbf{P}_3$, which were generated by models that have not encountered the classes within $\mathcal{C}_4$. Despite this, $\mathcal{M}_4$ is still able to identify unseen objects with high mAP, showcasing the impressive zero-shot ability of the RPN.

## C. Is the RoI Head robust to low-quality proposals?

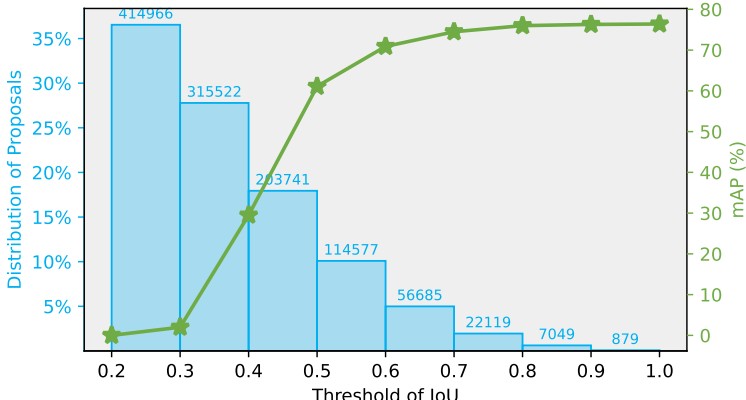

Figure 6: Plot: Results of $\mathcal{M}_{joint}$ after removing high-quality proposals with varying IoU threshold. Bar: The distribution of the proposals generated with $\mathcal{M}_{joint}$ over IoU. The number on the bar indicates the count of proposals.

A robust RoI head is capable of effectively offsetting RPN's forgetting. To evaluate the robustness of the RoI Head, we manually removed high-quality proposals during inference, *i.e.* high IoU with GTs, to assess the mAP result of the detector. As shown in Figure 6, when removing the proposals with IoU above 0.7, comparable final results can still be obtained (74.5% to 76.4%). In particular, the detector still manages to detect some instances and achieves noticeable results when removing proposals with IoU above 0.5, showing the strong robustness of the RoI Head. The robustness of the RoI Head can be attributed to the training process, where the RoI Head is trained to refine coarse proposals which have a very broad IoU range from a given value, 0.7 for example, to 1. The training with coarse proposals enables the RoI Head to refine rather low-quality proposals, leading to a robust performance of the RoI Head.

## D. Null Space Gradient Projection Details.

We introduced NSGP to alleviate the RoI feature shift caused by the evolution of the feature extractor. It is crucial for the NSGP to obtain a projection matrix that can project the gradient $G$ into the null space of the old example $\mathcal{X}_t = \{x_{t,i} \mid i \in \mathbb{N}, 1 \leq i \leq M_t\}$, where $M_t$ is the total number of inputs in the $t$-th training stages. An overview of NSGP are provided in Figure 7.

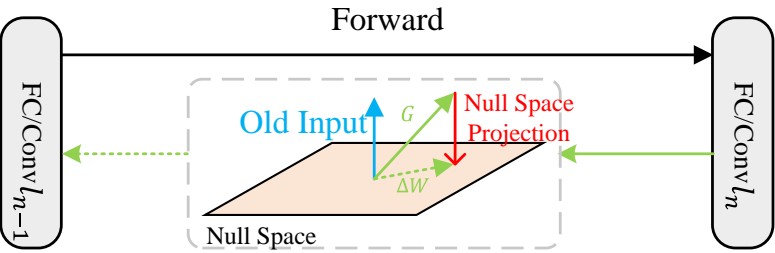

Figure 7: An overview of NSGP.

To obtain the projection matrix of an FC layer or a convolution layer with parameters $W$, we first compute the uncentered covariance of $\mathcal{X}_t$. Specifically, we can accumulate uncentered covariance matrix in $t$-th stage $\mathcal{T}_t$ as:

$$\mathcal{T}_t = \frac{1}{N_t - 1} \sum_{i=1}^{N_t} x_{t,i}^\top x_{t,i}. \tag{11}$$

After obtaining the uncentered covariance in $t$. The uncentered covariance of all previous training stages can be updated as

$$\bar{\mathcal{T}}_t = \frac{\bar{M}_{t-1}}{\bar{M}_t} \bar{\mathcal{T}}_{t-1} + \frac{M_t}{\bar{M}_t} \mathcal{T}_t. \tag{12}$$

Here, $\bar{M}_t = \bar{M}_{t-1} + M_t$. Then SVD is performed to obtain $U_t, \Lambda_t, (U_t)^\top$ as

$$U_t, \Lambda_t, (U_t)^\top = SVD(\bar{\mathcal{T}}_{t-1}) \tag{13}$$

Following (Lu et al.), we adaptively determine the nullity $R$ and retain $U_t'$ correspond to $R$ smallest diagonal singular vaules $\lambda$ in $\Lambda_t$. Finally, the projection matrix for $(t+1)$-th training stage is obtained by

$$\mathcal{B} = U_t'(U_t')^\top, \tag{14}$$

and the gradient $G$ is projected to the null space of $\mathcal{X}_t$ as

$$\Delta W = G\mathcal{B}. \tag{15}$$

It is a common practice in previous works (Wang et al., 2021; Lu et al.) to normalize the $\mathcal{B}$ as

$$\mathcal{B}' = \frac{\mathcal{B}}{||\mathcal{B}||_F}. \tag{16}$$

Unlike previous works, we adopt $\mathcal{B}$ as normalized $\mathcal{B}'$ will decrease the update stride of the model, leading to a slow and difficult optimization. The slow learner is beneficial to the classification task, as shown in SLCA (Zhang et al., 2023), but it is not applicable to components in Faster R-CNN except backbone. Thus we only apply $\mathcal{B}'$ to the backbone, adopting $\mathcal{B}$ to the rest of the components in the detector. EWC (Kirkpatrick et al., 2017) is adopted to regulate the update of parameterized normalization layers.

To justify our choice of only applying NSGP to the backbone and neck, we conduct experiments on all components of the detector, as show in Table 5. $\mathcal{B}$ or $\mathcal{B}'$ indicates that the gradient of component, except the backbone, is projected by $\mathcal{B}$ or $\mathcal{B}'$. Our experiments suggest that adopting NSGP in different components leads to results without significant fluctuations, suggesting the detector is not sensitive to the NSGP. Comparing $\mathcal{B}$ and $\mathcal{B}'$ suggests that lower scale for the update stride in neck, RPN and RoI Head leads to a significant decrease in performance. These results justify our choice of $\mathcal{B}$ instead of $\mathcal{B}'$.

Table 5: The experimental results of NSGP in different components with projection matrix $\mathcal{B}$ or $\mathcal{B}'$. The dataset we adopted is VOC (5-5).

|  | Backbone | +Neck | +RPN | +RoI Head |
|---|---|---|---|---|
| $\mathcal{B}'$ | 62.6 | 60.3 | 59.0 | 41.9 |
| $\mathcal{B}$ | 62.6 | **63.3** | 63.0 | 63.2 |

## E. Different strategy generating fine-grained prototypes.

To evaluate the effectiveness of the proposed fine-grained prototype generation process, we compared our method with clustering algorithms: K-Means and DBSCAN. To justify our choice of prototype instead of instances, we selected the center of the hypersphere instead of the averaging of the RoI features included in the hypersphere and named this method Instance. As shown in Table 6, our results outperform K-Means and DBSCAN by 0.7% on average, suggesting the effectiveness of the proposed method. Our prototypes surpass Instance by 1.4%, justifying the choice of the prototype instead of instances.

Table 6: Different strategy generating complementary prototypes of our method.

| Method | VOC(5-5) | | |
|---|---|---|---|
|  | 1-5 | 6-20 | 1-20 |
| K-Means | 62.7 | 65.6 | 64.9 |
| DBSCAN | 63.6 | 65.5 | 65.1 |
| Instance | 63.2 | 64.6 | 64.3 |
| Ours | **64.6** | **66.1** | **65.7** |

## F. RePRE Performance with Coarse Regional Prototype Only.

Our RePRE can surpass previous works even with only coarse prototype being replayed. We name NSGP-RePRE incorporated with coarse prototype only as the NSGP-RePRE-Coarse.

**PASCAL VOC Single-step Increments.** In Table 7, we make a comparison between our proposed method and existing approaches. NSGP-RePRE-Coarse surpasses previous state-of-the-art BPF by 1.7% in base classes and by 0.7% in all 20 classes, underscoring the effectiveness of our approach.

**PASCAL VOC Multi-step Increments.** The increases in initial classes indicate reduced forgetting with only coarse prototypes, while the improvements in 1-20 and the average reflect that our method achieves the optimal balance between stability and plasticity compared with previous methods. In Table 8, NSGP-RePRE-Coarse shows a 4.7% improvement over ABR in the initial classes in all 5 settings and a 2. 8% improvement in the 1-20 classes. Our method exceeds the performance of BPF by 4.5% in the base classes and 2.3% in the 1-20 classes.

**MS COCO Single Increments.** In Table 9, NSGP-RePRE-Coarse increased the average AP by 0.8% over the previous state-of-the-art in the 40-40 configuration. In the 70-10 scenario, the performance is close to that of joint training, with our method yielding results comparable to the previous SOTA BPF. These experimental results demonstrate the efficacy of NSGP-RePRE-Coarse.

Table 7: mAP@0.5 results on single incremental step on PASCAL VOC 2007. The best performance in each is presented with **bold**, and the second best is presented with underline.

| Method | 19-1 | | | | 15-5 | | | | 10-10 | | | | 5-15 | | | |
|---|---|---|---|---|---|---|---|---|---|---|---|---|---|---|---|---|
| | 1-19 | 20 | 1-20 | Avg | 1-15 | 16-20 | 1-20 | Avg | 1-10 | 11-20 | 1-20 | Avg | 1-5 | 5-15 | 1-20 | Avg |
| Joint | 76.4 | 76.4 | 76.4 | 76.4 | 78.3 | 70.7 | 76.4 | 74.5 | 76.9 | 76.0 | 76.4 | 76.4 | 73.6 | 77.4 | 76.4 | 75.5 |
| Fine-tuning | 12.0 | 62.8 | 14.5 | 37.4 | 14.2 | 59.2 | 25.4 | 36.7 | 9.5 | 62.5 | 36.0 | 36.0 | 6.9 | 63.1 | 49.1 | 35.0 |
| ORE (Joseph et al., 2021a) | 69.4 | 60.1 | 68.9 | 64.7 | 71.8 | 58.7 | 68.5 | 65.2 | 60.4 | 68.8 | 64.6 | 64.6 | - | - | - | - |
| OW-DETR (Gupta et al., 2022) | 70.2 | 62.0 | 69.8 | 66.1 | 72.2 | 59.8 | 69.1 | 66.0 | 63.5 | 67.9 | 65.7 | 65.7 | - | - | - | - |
| ILOD-Meta (Joseph et al., 2021b) | 70.9 | 57.6 | 70.2 | 64.2 | 71.7 | 55.9 | 67.8 | 63.8 | 68.4 | 64.3 | 66.3 | 66.3 | - | - | - | - |
| ABR (Yuyang et al., 2023) | 71.0 | **69.7** | 70.9 | 70.4 | 73.0 | **65.1** | 71.0 | 69.1 | 71.2 | 72.8 | 72.0 | 72.0 | 64.7 | 71.0 | 69.4 | 67.9 |
| FasterILOD (Ren et al., 2016) | 68.9 | 61.1 | 68.5 | 65.0 | 71.6 | 56.9 | 67.9 | 64.3 | 69.8 | 54.5 | 62.1 | 62.1 | 62.0 | 37.1 | 43.3 | 49.6 |
| PPAS (Zhou et al., 2020) | 70.5 | 53.0 | 69.2 | 61.8 | - | - | - | - | 63.5 | 60.0 | 61.8 | 61.8 | - | - | - | - |
| MVC (Yang et al., 2022) | 70.2 | 60.6 | 69.7 | 65.4 | 69.4 | 57.9 | 66.5 | 63.7 | 66.2 | 66.0 | 66.1 | 66.1 | - | - | - | - |
| PROB (Zohar et al., 2023) | 73.9 | 48.5 | 72.6 | 61.5 | 73.5 | 60.8 | 70.1 | 67.0 | 66.0 | 67.2 | 66.5 | 66.5 | - | - | - | - |
| PseudoRM (Yang et al., 2023) | 72.9 | 67.3 | 72.6 | 70.1 | 73.4 | 60.9 | 70.3 | 66.9 | 69.1 | 68.6 | 68.9 | 68.9 | - | - | - | - |
| MMA (Cermelli et al., 2022) | 71.1 | 63.4 | 70.7 | 67.2 | 73.0 | 60.5 | 69.9 | 66.7 | 69.3 | 63.9 | 66.6 | 66.6 | 66.8 | 57.2 | 59.6 | 62.0 |
| BPF (Mo et al., 2024) | 74.5 | 65.3 | 74.1 | 69.9 | 75.9 | 63.0 | 72.7 | 69.5 | 71.7 | **74.0** | 72.9 | 72.9 | 66.4 | **75.3** | 73.0 | 70.9 |
| NSGP-RePRE-Coarse | 76.2 | 66.5 | 75.8 | 71.4 | 77.1 | 62.0 | 73.4 | 69.6 | 73.7 | 73.2 | 73.4 | 73.5 | 68.4 | 74.5 | **73.0** | **71.5** |
| NSGP-RePRE | 76.3 | 69.0 | **76.0** | **72.7** | **77.5** | 61.8 | **73.6** | **69.7** | **75.3** | 72.7 | **74.0** | **74.0** | **68.5** | 74.5 | **73.0** | **71.5** |

Table 8: mAP@0.5 results on multiple incremental steps on PASCAL VOC 2007. The best performance in each is presented with **bold**, and the second best is presented with underline.

| Method | 10-5(3tasks) | | | 5-5(4tasks) | | | 10-2(6tasks) | | | 15-1(6tasks) | | | 10-1(11tasks) | | |
|---|---|---|---|---|---|---|---|---|---|---|---|---|---|---|---|
| | 1-10 | 11-20 | 1-20 | 1-5 | 6-20 | 1-20 | 1-10 | 11-20 | 1-20 | 1-15 | 16-20 | 1-20 | 1-10 | 11-20 | 1-20 |
| Joint | 76.9 | 76.0 | 76.4 | 73.6 | 77.4 | 76.4 | 76.9 | 76.0 | 76.4 | 78.3 | 70.7 | 76.4 | 76.9 | 76.0 | 76.4 |
| Fine-tuning | 5.3 | 30.6 | 18.0 | 0.5 | 18.3 | 13.8 | 3.8 | 13.6 | 8.7 | 0.0 | 10.5 | 5.3 | 0.0 | 5.1 | 2.6 |
| ABR (Yuyang et al., 2023) | 68.7 | 67.1 | 67.9 | 64.7 | 56.4 | 58.4 | 67.0 | 58.1 | 62.6 | 68.7 | **56.7** | 65.7 | 62.0 | **55.7** | 58.9 |
| FasterILOD (Ren et al., 2016) | 68.3 | 57.9 | 63.1 | 55.7 | 16.0 | 25.9 | 64.2 | 48.6 | 56.4 | 66.9 | 44.5 | 61.3 | 52.9 | 41.5 | 47.2 |
| MMA (Cermelli et al., 2022) | 66.7 | 61.8 | 64.2 | 62.3 | 31.2 | 38.9 | 65.0 | 53.1 | 59.1 | 68.3 | 54.3 | 64.1 | 59.2 | 48.3 | 53.8 |
| BPF (Mo et al., 2024) | 69.1 | **68.2** | 68.7 | 60.6 | 63.1 | 62.5 | 68.7 | 56.3 | 62.5 | 71.5 | 53.1 | 66.9 | 62.2 | 48.3 | 55.2 |
| NSGP-RePRE-Coarse | 71.9 | 66.2 | 69.1 | 65.9 | 63.8 | 64.3 | 68.7 | 54.8 | 61.8 | 77.0 | 53.9 | 71.2 | **71.2** | 50.6 | 60.9 |
| NSGP-RePRE | **72.4** | 67.6 | **70.0** | 64.6 | **66.1** | **65.7** | **70.1** | **58.8** | **64.4** | **77.7** | 55.0 | **72.0** | 69.9 | 55.1 | **62.5** |

Table 9: mAP results on MS COCO 2017 at different IoU. The best performance in each is presented with **bold**, and the second best is presented with underline.

| Method | 40-40 | | | 70-10 | | |
|---|---|---|---|---|---|---|
| | AP | AP50 | AP75 | AP | AP50 | AP75 |
| Joint | 36.7 | 57.8 | 39.8 | 36.7 | 57.8 | 39.8 |
| Fine-tuning | 19.0 | 31.2 | 20.4 | 5.6 | 8.6 | 6.2 |
| ILOD-Meta (Joseph et al., 2021b) | 23.8 | 40.5 | 24.4 | - | - | - |
| ABR (Yuyang et al., 2023) | 34.5 | **57.8** | 35.2 | 31.1 | 52.9 | 32.7 |
| FasterILOD (Ren et al., 2016) | 20.6 | 40.1 | - | 21.3 | 39.9 | |
| PseudoRM (Yang et al., 2023) | 25.3 | 44.4 | - | - | - | - |
| MMA (Cermelli et al., 2022) | 33.0 | 56.6 | 34.6 | 30.2 | 52.1 | 31.5 |
| BPF (Mo et al., 2024) | 34.4 | 54.3 | 37.3 | 36.2 | **56.8** | 38.9 |
| NSGP-RePRE-Coarse | 35.2 | 55.3 | 38.1 | 36.3 | 55.8 | 39.6 |
| NSGP-RePRE | **35.4** | 55.3 | **38.6** | **36.5** | 56.0 | **39.8** |

