# OpenReview forum: "Demystifying Catastrophic Forgetting in Two-Stage Incremental Object Detector"
_ICML.cc/2025/Conference — ICML 2025 poster_

### Official Review · Reviewer_K17L · 2025-03-10

**Overall Recommendation:** 3

**Summary:**

The paper focuses on catastrophic forgetting in the two-stage object detector. The authors first analyse the forgetting in component-level and reveal that RoI Head classifier is the primary cause of catastrophic forgetting. Then the authors propose Regional Prototype Replay (RePRE) to mitigate forgetting via replay of coarse prototypes and fine-grained prototypes. The authors also propose using Null Space Gradient Projection (NSGP) to eliminate prototype-feature misalignment. Experiments on VOC and COCO show that the proposed method, NSGP-RePRE, can significantly improve the performance of Faster-RCNN in IOD.

## update after rebuttal

After reviewing all the reviews and responses, my primary concerns have been addressed. At this point, I am inclined to accept.

**Claims And Evidence:**

No.
The authors claim that 'In sequential tasks, the stability of the RPN recall ability is largely maintained.'. However, all experiments are conducted on VOC and COCO, lacking experiments on drastic variations in target scale or target aspect ratio between different tasks.

**Essential References Not Discussed:**

No.

**Experimental Designs Or Analyses:**

No. The paper lacks experiments on more datasets. Conclusions drawn from COCO and VOC may not be universally applicable.

**Methods And Evaluation Criteria:**

No.

**Other Comments Or Suggestions:**

1. A Grammatical error in line 258~259 should be corrected: "To capture the entire spectrum of useful information on the distribution of RoI features."

2. The citation of BPF (ECCV'24) should be a conference version.

3. The result of ABR in the last column of Tab 2 is incorrectly bolded.

**Other Strengths And Weaknesses:**

## Strengths

1. The paper identifies a meaningful perspective in IOD with two-stage detectors and propose a solution to mitigate forgetting in classification.

2. Experiments on two widely-used datasets are adequate. The proposed method reaches state-of-the-art on multiple datasets and settings.

3. The writing of the papre is easy to follow and it is clearly structured.

## Weakness

**Major**

1. The paper lacks experiments on more datasets with significant variations in target scale and aspect ratio, which makes the conclusions of the paper more limited.

2. The NSGP seems to be a simple application of previous method, which is less innotative.

**Minor**

1. The authors only use Faster R-CNN in experiments. It is suggested to try more two-stage detectors like Cascade-RCNN to verify the generalizability of the conclusions to two-stage detectors.

2. The ablation in Tab 4 is insufficient. The authors are suggested to conduct more experiments on other settings, such as VOC 10-10.

3. The paper lacks visualization results. Including visualization results that demonstrate how the proposed method corrects misclassifications would provide readers with a more concrete understanding of its performance.

**Questions For Authors:**

1. All experiments are conducted on VOC and COCO. Do the findings about catastrophic forgetting in Faster R-CNN still hold on datasets with significant variations in target scale and aspect ratio (such as remote sensing detection datasets)?
2. The author mentions that the baseline employs a pseudo-labeling strategy, and the reviewer is interested in the performance of the proposed method after removing the pseudo-labels.

**Relation To Broader Scientific Literature:**

Previous works in the IOD field usually treats the detector as a whole, lacking a fine-grained analysis. This paper decouples localization and classification and analyzes catastrophic forgetting with experimental results in two-stage detectors at the component level.

**Theoretical Claims:**

There is no Theoretical Claim in the paper.

---

> ### Author Rebuttal · Authors · 2025-04-01
>
> Thanks for the insightful comments.
>
> **Q1:** Dataset concern.
>
> **R1:** To show the generalizability of our key findings, we conducted experiments on a wildly used remote sensing detection datasets DIOR. **The three key findings still hold with different two-stage IODs.** As shown in this link [DIOR](https://anonymous.4open.science/r/aosdhoaihfoiahsodjasjiohf/d.pdf), all curves are aligned with VOC dataset. Our results show that the detector's RPN struggles with unseen data, as indicated by the red curves. However, the detector performs well on seen data, suggesting the RPN and the RoI Head's regressor remain resilient and do not forget previously learned knowledge on DIOR.
>
> We also conducted experiments on DIOR with our framework, as shown in
>
> |DIOR|5-5|||\||10-10|||
> |-|-|-|-|-|-|-|-|
> ||1-5|6-20|1-20|\||1-10|11-20|1-20|
> |Baseline|43.6|56.9|53.6|\||61.03|63.9|62.5|
> |NSGP-RePRE|54.8|57.0|56.5|\||66.4|62.25|64.3|
>
> Our framework surpass the baseline by 2.9\% in 5-5 setting and 1.8\% in 10-10 setting, further showing the effectiveness of framework.
>
> **Q2:** NSGP's novelty concern.
>
> **R2:**  Although NSGP has been explored in incremental classification, it is non-trivial to apply NSGP in IOD. The following table demonstrates the performance applying NSGP from Backbone to RoI Head accumulatively under VOC 5-5 setting.
>
> ||Backbone|+FPN|+RPN|+RoIHead|Ours|
> |-|-|-|-|-|-|
> |NSGPonly|62.6|63.3|63|63.2|65.7|
>
> The catastrophic forgetting in two-stage detectors is mainly caused by the severe classifier instability in the RoI Head. However, directly appling NSGP on these two-stage object detector (i.e., FPN, RPN,RoIHead)  shows limited performance improvements and can not well address the classifier instability issue, as illustrated in the above Table.
>
> Instead, the proposed Regional Prototype Replay (RePRE) module addresses  this issue by replaying coarse and fine-grained region prototypes in the RoI Head's classification branch. NSGP serves as an assistant component in our framework by mitigating semantic drift caused by parameter updates, thereby preventing toxic replay in RePRE. As shown in the table, our RePRE achieves a +2.4% gain compared with +FPN, underscoring the effectiveness of our proposed framework.
>
> **Q3:** Architecutre-specific concern.
>
> **R3:** To show the generalizability of our key findings, we also conducted experiments with two popular  two-stage detectors, i.e. Cascade-RCNN and Vanilla Faster RCNN. **The three key findings still hold with different two-stage IODs as shown in these links [CascadeRCNN](https://anonymous.4open.science/r/aosdhoaihfoiahsodjasjiohf/c.pdf) and [VanillaFasterRCNN](https://anonymous.4open.science/r/aosdhoaihfoiahsodjasjiohf/v.pdf).** Detailed discussion can also be found in our response to reviewer huSY R3.
>
> **Q4:** Insufficient ablation concern.
>
> **R4:**  Thanks for the suggestion. We conducted ablation study in VOC 10-10 setting as shown in
>
> ||||VOC|(10-10)||
> |-|-|-|-|-|-|
> |NSGP|Coarse|Fine|1-10|11-20|1-20|
> ||||69.3|73.3|71.3|
> |x|||71.8|73.2|72.5|
> ||x||70.5|73.8|72.1|
> |x|x||73.7|73.2|73.4|
> |x|x|x|75.3|72.7|74.0|
>
> The same conclusion can be drawn from this table as in ablation study in 5-5 setting.
>
> **Q5:** Visualization of the results.
>
> **R5:** As shown in this link [visualization](https://anonymous.4open.science/r/aosdhoaihfoiahsodjasjiohf/vis.pdf), we visualized images from VOC2007 test set under 10-10 setting. Task 1 represents visualization with model from time step 1 while baseline and NSGP-RePRE represents the visualization results are from time step 2 trained with corresponding strategy. In (a) baseline forgets "boat". In (b) and (c), baseline forgets "cat" and "car" due to the interference of new classes "dog" and "motorbike". Our NSGP-RePRE successfully remembers old classes while learning new classes effectively, suggesting that our methods achieves better stability while retaining comparable plasticity compared with baseline.
>
> **Q6:** The pseudo labeling concerns
>
> **R6:** One major problem in IOD is that objects from past tasks can be included in the subsequent tasks yet their labels are not annotated. For example, airplanes will be labeled as "airplane" in the first task but considered as background in the subsequent tasks. Optimizing with wrong label will leads to a drastic performance drop. Pseudo labeling(Mo et al.,2024; Liu et al.,2023;) are wildly adopted to alleviate such performance drop. Thus, we also choose pseudo labeling as our baseline to avoid this problem.
>
> We also conducted experiments under the setting of without pseudo label. As shown in
>
> |W/o Pseudo Label|\||5-5|||\||10-10|||
> |-|-|-|-|-|-|-|-|-|
> ||\||1-5|6-20|1-20|\||1-10|11-20|1-20|
> |Baseline (w/o pseudo label)|\||0|28.2|21.2|\||14.5|66.9|40.7|
> |NSGP-RePRE (w/o pseudo label) |\||50.5|47.8|48.5|\||66.2|58.9|64.3|
>
> Our NSGP-RePRE achieves +20% performance gain compared with baseline, demonstrating the effectiveness of our framework.

---

> > ### Comment · Reviewer_K17L · 2025-04-02
> >
> > Thanks for the responses. It resolved my main concerns.

---

> > > ### Author Response · Authors · 2025-04-08
> > >
> > > We sincerely thank the reviewer for their valuable comments and constructive suggestions. We truly appreciate the time and effort you dedicated to reviewing our submission and the opportunity to clarify and improve our work.

---

### Official Review · Reviewer_5Yg2 · 2025-03-12

**Overall Recommendation:** 4

**Summary:**

The paper addresses the critical challenge of catastrophic forgetting in incremental object detection (IOD). The authors focus on the Faster R-CNN architecture and identify that catastrophic forgetting predominantly occurs in the RoI Head classifier, while the regressor remains robust across incremental stages. Based on this insight, they propose NSGP-RePRE, which combines Regional Prototype Replay (RePRE) and Null Space Gradient Projection (NSGP) to mitigate forgetting in the RoI Head classifier. The method achieves state-of-the-art performance on the Pascal VOC and MS COCO datasets under various incremental learning settings.

**Claims And Evidence:**

Yes

**Essential References Not Discussed:**

NA

**Experimental Designs Or Analyses:**

Sound

**Methods And Evaluation Criteria:**

Yes

**Other Comments Or Suggestions:**

See the Strengths And Weaknesses

**Other Strengths And Weaknesses:**

Strengths:

1. The paper provides a significant insight into the nature of catastrophic forgetting in two-stage object detectors, specifically identifying the RoI Head classifier as the primary source of forgetting. This challenges conventional assumptions and offers a new direction for addressing forgetting in IOD.

2. Comprehensive experimental results.

3. Topic is interesting

Weaknesses:

1. More theorical analysis, such as complexity analysis can be provided.

**Questions For Authors:**

See the Strengths And Weaknesses

**Relation To Broader Scientific Literature:**

NA

**Theoretical Claims:**

NA

---

> ### Author Rebuttal · Authors · 2025-03-31
>
> We sincerely appreciate the reviewer's positvie feedback and insightful comment. The Questions and Responses are as follows.
>
> **Q1:** More theorical analysis, such as complexity analysis can be provided.
>
> **R1:** Thanks for the suggestions. We provide a comprehensive analysis of the parameter, computational, and memory complexity of NSGP-RePRE:
>
> **Parameter Analysis:**
>
> The origional detectors contains a total of 96.89 M trainable parameters. Our approach maintains the same parameter count, ensuring no increase in model complexity.
>
> **Computational Complexity:**
>
> To assess the computational complexity during training, we present the FLOPs for key components of the model, as shown in the table below:
>
> |       | forward | +RePRE     | backward | +NSGP    |
> |-------|:---------:|:-----------------:|:----------:|:----------:|
> | GFLOPs | 551.35   | +2.78 | 1102.7    | +118.1 |
>
> It is important to note that the computational complexities of RePRE and NSGP do not scale with batch size, whereas the forward and backward passes of the detector do. As shown in the table, the majority of the computational cost during training arises from the detector's forward and backward passes. In contrast, RePRE and NSGP contribute only about 1% of the additional computation compared to the forward and backward operations when scaling batch size to commonly used 8.
>
> We also present the actual training time for different components on 1 RTX 3090 GPUs in the following table:
>
> |           | Baseline | NSGP   | NSGP+RePRE |
> |-----------|----------|--------|------------|
> | Time/iter | 0.714s   | 0.719s | 0.720s     |
>
> This table suggests that the additional computational cost of NSGP-RePRE is minimal, adding only ~1% overhead per iteration compared to baseline training, which is aligned with our complexity analysis.
>
>
> **Memory cost of RePRE:**
>
> The memory footprint of our RePRE scales linearly with the increase of the number of classes. In our implementation, each class consumes approximately 3.8Mb of memory in float32 with each prototype consuming 0.38Mb. Our method achieves performance comparable to the previous example-based SOTA method, ABR, using only one coarse prototype per class and without relying on NSGP in the 10-10 setting. Additionally, it consumes about 25% of the memory required by ABR. As shown in
>
> | VOC(10-10)   |         |       |       |       |
> |--------------|---------|-------|-------|-------|
> | Type         | Memory↓ | 1-10  | 11-20 | 1-20  |
> | ABR          | 15.5Mb  | 71.2  | 72.3  | 72.0  |
> | RePRE-Coarse | 3.8Mb   | 70.5  | 73.8  | 72.1  |
> | NSGP-RePRE   | 38Mb    | 75.3  | 72.7  | 74.0  |

---

### Official Review · Reviewer_huSY · 2025-03-12

**Overall Recommendation:** 3

**Summary:**

The paper investigates catastrophic forgetting in incremental object detection using standard Faster-RCNN architecture. The authors show that catastrophic forgetting mainly happens in the RoI part of the model, and the regressor behaves more robustly while learning subsequent tasks. Based on their observations, the authors propose Regional Prototype Replay (RePRE) method, which mitigates classifier forgetting via replay of coarse and fine-grained prototypes and Null Space Gradient Projection (NSGP). NSGPRePRE is evaluated on the Pascal VOC and MS COCO datasets, where it demonstrates improved stability compared to the other IOD methods.

**Claims And Evidence:**

The claims made in Section 3 are consistent with the experimental results.

However, the claim that the Avg metric reflects a better trade-off between stability and plasticity (Line 371) is problematic, and the authors do not provide a good justification for it. For example, separately evaluating the accuracy of base and novel classes after each incremental learning step would offer clearer insights into how the method balances stability and plasticity over time.

**Essential References Not Discussed:**

See **Questions**.

**Experimental Designs Or Analyses:**

See **Questions**.

**Methods And Evaluation Criteria:**

The proposed evaluation is well-suited for the problem.

**Other Comments Or Suggestions:**

See **Questions**.

**Other Strengths And Weaknesses:**

See **Questions**.

**Questions For Authors:**

1. NSGP projects gradients into the null space of old tasks to prevent feature drift. Could this restrict the model's plasticity, especially when new tasks require significant feature adaptation? How does NSGP balance stability and plasticity?

2. RePRE requires storing multiple prototypes per class. How does the memory footprint scale with the number of incremental stages? Is there a risk of prototype redundancy or interference when handling highly similar classes?

3. The conclusion about catastrophic forgetting being localized to the RoI Head classifier is based solely on Faster R-CNN. Have the authors validated this finding on other two-stage architectures? If not, how can we ensure this is a generalizable insight rather than architecture-specific?

4. How critical is pseudo-labeling to NSGP-RePRE’s performance?

5. How does the NSGP affect training time compared to baseline methods? Is the method practical for real-time applications?

6. As shown in the results, the proposed method performs well on base classes. However, I noticed that in Tables 1, 2, and 3, the proposed method underperforms its counterparts on incremental tasks. Does this imply that the method overly prioritizes stability while exhibiting weaker plasticity for learning new tasks?

While the authors present a novel finding, their validation is limited to Faster R-CNN, with little evidence of broader applicability. Additionally, critical experimental validations are still missing. I am temporarily giving this paper a weak reject, but I will continue to follow the author's response and the comments from other reviewers.

**Relation To Broader Scientific Literature:**

The key contributions of this paper are rooted in and extend the broader literature on incremental learning and object detection, addressing critical gaps in prior work. Previous IOD methods do not dissect component-level contributions to forgetting, and this paper reveals classifier instability as a key source of forgetting in IOD.

**Theoretical Claims:**

Not applicable.

---

> ### Author Rebuttal · Authors · 2025-04-01
>
> Thanks for the insightful comments.
>
> **Q0: On the Avg metric.**
>
> **R0:** We show the Avg performance at every step.
>
> |10-10|Step1|\||Step 2|Base|New|Avg|All|
> |-|-|-|-|-|-|-|-|
> |Baseline|77.8|\||Baseline|69.3|73.3|71.3|71.3|
> |BPF*|77.8|\||BPF*|71.8|73.4|72.6|72.6|
> |NSGP-RePRE|77.8|\||NSGP-RePRE|75.3|72.7|74|74|
>
> |5-5|Step1|\||Step2|Base|New|Avg|All|\||Step3|Base|New|Avg|All|\||Step4|Base|New|Avg|All|
> |-|-|-|-|-|-|-|-|-|-|-|-|-|-|-|-|-|-|-|-|
> |Baseline|77.4|\||Baseline|63.1|79.9|71.5|71.5|\||Baseline|64.7|76.1|70.4|68.5|\||Baseline|58.0|59.6|58.8|58.4|
> |NSGP-RePRE|77.4|\||NSGP-RePRE|72.6|78.1|75.3|75.3|\||NSGP-RePRE|70.8|76.3|73.6|72.6|\||NSGP-RePRE|67.9|59.0|63.5|65.7|
>
>
> All represents the mAP of all seen classes, e.g. 1-15 in 5-5 setting step 3. Our method achieves the best Avg and All mAP at every time step, showing its superior balanced stability-plasticity properties.
>
>
> **Q1 & Q6:On the plasticity-stability trade-off.**
>
> **R1:** The plasticity-stability trade-off is explicitly controlled by the nullity of the uncentered feature covariance. In our experiments, we adopt an adaptive approach VPT-NSP^2(Lu e al.,2024)
> and achieved better trade-off.
>
> **Q2: On the scaling of memory requirement and prototype interference risk.**
>
> **R2:** The memory footprint of our RePRE scales linearly with increasing number of classes. Each class consumes approximately 3.8Mb, and each prototype consumes 0.38Mb.
> The comparison between our method and previous exampler-based SOTA method ABR is shown below.
>
> |VOC(10-10)|||||
> |-|-|-|-|-|
> |Type|Memory↓|1-10|11-20|1-20|
> |ABR|15.5Mb|71.2|72.3|72.0|
> |RePRE-Coarse|3.8Mb|70.5|73.8|72.1|
> |NSGP-RePRE|38Mb|75.3|72.7|74.0|
>
> We address redundancy by setting a distance between prototypes to capture the whole distribution of the feature space. Our results show consistent gains, suggesting effective handling of similar classes.
>
> **Q3: conclusion on other two-stage architectures.**
>
> **R3:** To show the generalizability of our key findings, we also conducted experiments with two popular  two-stage detectors, i.e. Cascade-RCNN and Vanilla Faster RCNN without FPN and RoI Align. **The three key findings still hold with different two-stage IODs,** as shown in these links [CascadeRCNN](https://anonymous.4open.science/r/aosdhoaihfoiahsodjasjiohf/c.pdf) and [VanillaFasterRCNN](https://anonymous.4open.science/r/aosdhoaihfoiahsodjasjiohf/v.pdf). We conducted the same anotomy to these detectors in our paper, all curves are aligned with Faster RCNN using Pascal VOC 5-5. These two detectors' RPN Recall curve show that seen models even can surpass current model, suggesting that our findings are universal in most two-stage IODs.
>
> We also evaluated our method on these detectors.
>
>
> |CascadeRCNN|5-5|||\||10-10|||
> |-|-|-|-|-|-|-|-|
> ||1-5|6-20|1-20|\||1-10|11-20|1-20|
> |Baseline|57.2|65.4|63.4|\||69.7|74.3|72.0|
> |NSGP-RePRE|66.8|66.6|66.7|\||74.1|74.6|74.4|
>
>
> |VanillaFasterRCNN|5-5|||\||10-10|||
> |-|-|-|-|-|-|-|-|
> ||1-5|6-20|1-20|\||1-10|11-20|1-20|
> |Baseline|13.3|27.5|23.9|\||27.2|32.3|29.8|
> |NSGP-RePRE|19.2|26.9|25.0|\||29.8|32.6|31.2|
>
> Our method achieved noticeable performance improvement.
>
> **Q4: Concerns about pseudo-labeling.**
>
> **R4:** One major problem in IOD is that objects from past tasks can be included in the subsequent tasks yet their labels are not annotated. Optimizing with wrong label will lead to a drastic performance drop. Pseudo labeling(Mo et al.,2024; Liu et al.,2023;) are widely adopted to alleviate such performance drop. Thus, we also choose pseudo labeling as our baseline.
>
> We also conducted experiments with pseudo label been ignored(neither foreground nor background). As shown in
>
> |W/o Pseudo Label|\||5-5|||\||10-10|||
> |-|-|-|-|-|-|-|-|-|
> ||\||1-5|6-20|1-20|\||1-10|11-20|1-20|
> |Baseline (w/o pseudo label)|\||0|28.2|21.2|\||14.5|66.9|40.7|
> |NSGP-RePRE (w/o pseudo label) |\||50.5|47.8|48.5|\||66.2|58.9|64.3|
>
> Our NSGP-RePRE achieves +20% performance gain compared with baseline under the settings of without pseudo label module, demonstrating the effectiveness of our framework.
>
> **Q5: How NSGP affect training time.**
>
> **R5:** Our NSGP-RePRE maintains high efficiency.
> NSGP introduces additional training time by SVD decomposition and null-space projection. Both SVD and projection's required time only increases with the size of the model. For SVD, it only computes once at every incremental step and only took around 30 seconds in our experiments, much less than the whole  training time. As for null-space projection, it also adds only ~1% overhead per iteration compared to baseline training as shown in
>
> ||Baseline|NSGP|NSGP+RePRE|
> |-|-|-|-|
> |Time/iter|0.714s|0.719s|0.720s|
>
> NSGP-RePRE does not introduces extra overhead during inference, which makes it practical for real-time application.

---

> > ### Comment · Reviewer_huSY · 2025-04-03
> >
> > Thank you for the response. Most of my concerns were resolved at this point.
> >
> > Q0: BPF is still missing from the comparison for the 5-5 setting, while the results for the 10-10 setting show that it is competitive and outperforms NSGP-RePRE in terms of the performance on the new classes.
> >
> > Q1&6: I would encourage including a discussion on this in the manuscript, instead of merely referring to another paper, as this is an interesting aspect of your approach.
> >
> > Given the author's response and other reviews, at this point, I am also leaning toward acceptance of the paper.

---

> > > ### Author Response · Authors · 2025-04-08
> > >
> > > Thank you for your comment.
> > >
> > > **Q0:** The original BPF paper does not provide source code for long-sequence IOD, nor does it report intermediate results at various learning stages. We are currently re-implementing BPF in the long-sequence setting, and we will update its performance results in our project page once we have a reliable and consistent implementation.
> > >
> > > Regarding the lower performance on the "New" class, it's important to highlight that continual learning focuses on achieving a balance between *stability* (retaining previous knowledge) and *plasticity* (acquiring new knowledge), rather than optimizing performance on newly introduced classes alone. While our method shows slightly lower accuracy on "New" in the 10-10 setting, it achieves superior results on the Avg and All metrics—both of which are better indicators of the stability-plasticity trade-off. This suggests that our method performs better overall in the IOD task compared to competing methods.
> > >
> > > **Q1 & Q6:** Thank you for your suggestion. We will consider incorporating it into the final version of the paper.
> > >
> > > The stability-plasticity trade-off in our method is explicitly controlled through the nullity of the uncentered feature covariance matrix. We adopt the adaptive nullity selection approach proposed in VPT-NSP² (Lu et al., 2024), which dynamically determines nullity during training. To evaluate its effectiveness, we conduct experiments using different eigenvalue thresholds defined as $\beta \times \lambda_{min}$, where $\lambda$ denotes the eigenvalues.
> > >
> > > The following table shows the mAP results under different β values in the 5-5 setting:
> > >
> > > | β      | 10   | 30   | 50   | 70   | 90   | 100  | Adaptive |
> > > |--------|------|------|------|------|------|------|----------|
> > > | mAP    | 61.3 | 63.2 | 63.7 | 64.6 | 65.1 | 64.3 | **65.7** |
> > >
> > > This result demonstrates that adaptive nullity achieves the best mAP, indicating that it provides the most effective balance between plasticity and stability in IOD.
> > >
> > > We sincerely thank the reviewer for your valuable comments and constructive suggestions. Your feedback has been instrumental in helping us improve the quality of our work.

---

### Official Review · Reviewer_2Xwn · 2025-03-13

**Overall Recommendation:** 4

**Summary:**

This paper addresses the challenge of catastrophic forgetting in incremental object detection, particularly in two-stage detectors like Faster R-CNN. The authors identify that catastrophic forgetting predominantly occurs in the RoI Head classifier, while the RPN and regression branches remain robust across incremental stages. Based on these findings, they propose NSGP-RePRE, a framework combining Regional Prototype Replay (RePRE) to mitigate classifier forgetting via coarse and fine-grained prototypes and Null Space Gradient Projection (NSGP) to counteract feature extractor drift by projecting gradients orthogonally to the subspace of old task inputs. This approach ensures alignment between prototypes and updated feature distributions. The experiments are conducted on the PASCAL VOC and MS COCO datasets under various incremental learning settings.

**Claims And Evidence:**

Yes, the two components, RePRE and NSGP are well described, and the motivation is clear.

**Essential References Not Discussed:**

No

**Experimental Designs Or Analyses:**

The experimental design follows previous works and the analysis is sound.

**Methods And Evaluation Criteria:**

Yes, the paper adopted standard evaluation criteria as in the literature.

**Other Comments Or Suggestions:**

no

**Other Strengths And Weaknesses:**

Strengths:
1. Insights into Catastrophic Forgetting in IOD
2. Extensive experiments and good results
3. Clear writing.

Weaknesses:
1. Using NSGP has been explored in the literature for continual learning.
2. Although the specific form of replying regional prototype has not been explored, the idea of prototype based continual learning has been proposed in the literature.
Online Prototype Learning for Online Continual Learning, ICCV 2023
Prototype-Guided Memory Replay for Continual Learning, IEEE TNNLS 2024

The above two points make the paper's contribution is not so significant.


=================  After rebuttal ================
Based on the authors' feedback and other reviews, I would like to upgrade my score from weak accept to accept.

**Questions For Authors:**

no

**Relation To Broader Scientific Literature:**

By addressing the unique challenges of IOD, RePRE contributes to the broader literature on replay-based incremental learning. NSGP focuses on null-space projections for incremental learning, whose core idea has been explored in “Training networks in null space of feature covariance for continual learning, CVPR 2021”

**Theoretical Claims:**

There is no fundamental theoretical claim.

---

> ### Author Rebuttal · Authors · 2025-03-31
>
> We sincerely appreciate the reviewer's positvie feedback and insightful comment. The Questions and Responses are as follows.
>
> **Q1**: Using NSGP has been explored in the literature for continual learning.
>
> **R1**:Although NSGP has been explored in incremental classification, it is non-trivial to apply NSGP in IOD. The following table demonstrates the performance applying NSGP from Backbone to RoI Head accumulatively under VOC 5-5 setting.
>
> |          | Backbone | +FPN   | +RPN  |  +RoIHead | Ours |
> |----------|----------|--------|-------|-----------|------|
> | NSGP only| 62.6     | 63.3   | 63    | 63.2      | 65.7 |
>
> The catastrophic forgetting in two-stage detectors is mainly caused by the severe classifier instability in the RoI Head. However, directly appling NSGP on these two-stage object detector (i.e., FPN, RPN,RoIHead)  shows limited performance improvements and can not well address the classifier instability issue, as illustrated in the above Table.
>
> Instead, the proposed Regional Prototype Replay (RePRE) module addresses  this issue by replaying coarse and fine-grained region prototypes in the RoI Head's classification branch. NSGP serves as an assistant component in our framework by mitigating semantic drift caused by parameter updates, thereby preventing toxic replay in RePRE. As shown in the table, our RePRE achieves a +2.4% gain compared with +FPN, underscoring the effectiveness of our proposed framework.
>
> **Q2**: Although the specific form of replying regional prototype has not been explored, the idea of prototype based continual learning has been proposed in the literature. Online Prototype Learning for Online Continual Learning, ICCV 2023 Prototype-Guided Memory Replay for Continual Learning, IEEE TNNLS 2024
>
> **R2**: While naively applying prototype-based methods on the classifier does achieves performance gain for the detector, our work makes distinct contributions in the context of IOD as we considered RoI Head's uniqueness in pre-processing MLPs before classifier. Further, we extend our method to fine-grained regional prototype replay to capture the distribution of regional object features, which is crucial for preserving old knowledge in continual learning.
>
> To validate our design, we compare NSGP-RePRE against a baseline ("Classifier") that applies prototype replay only to the classifier (mimicking classification-focused approaches):
>
> |                  | 5-5  |      |      |\|| 10-10 |       |      |
> |------------------|------|------|------|-------|-------|------|----|
> |                  | 1-5  | 6-20 | 1-20 |\|| 1-10  | 11-20 | 1-20 |
> | w/o Prototype    | 62.3 | 63.6 | 63.3 |\|| 71.8  | 73.2  | 72.5 |
> | Classifier       | 63.6 | 63.3 | 63.4 |\|| 73.2  | 73.3  | 73.2 |
> | NSGP-RePRE       | 64.6 | 66.1 | 65.7 |\|| 75.3  | 72.7  | 74.0 |
>
> As shown in the table, our method outperforms "w/o Prototype" by +2.4%(5-5) and +1.5%(10-10). Though Classifier achieves performance gain compared with w/o Prototype, our method outperforms "Classifier" by +2.3% (5-5) and +0.8% (10-10). These results demonstrating that regulating only the classifier (as in classification-based works) is insufficient for IOD.
>
> **In addition, we also find our finding still holds across two extra two-stage detectors and a remote sensing dataset, which further highlights the reliability of our findings and the effectiveness of our framework. The reviewer may find them in our response to Reviewer huSY(Q3, architecture) and Reviewer K17L(Q1,dataset).**
>
> We emphasize that our goal is not to propose an overly complex method, but to offer a simple yet effective solution grounded in our analysis of catastrophic forgetting in two-stage detectors. We hope this work contributes to bridging the gap between continual learning for classification and detection.
>
> In the end, we like to thank the reviewer for their valuable suggestions and questions.

---

### Decision · Program_Chairs · 2025-05-01

**Decision:**

Accept (poster)

**Comment:**

The reviewers highlighted this paper's novel setting, good writing, and convincing experiments. The authors also provided reasonable answers to the doubts about the details of the proposed dataset and method. The reviewers’ concerns were well-addressed. Therefore, the decision is to recommend Acceptance.